# A Wideband High Gain Differential Patch Antenna Featuring In-Phase Radiating Apertures

**DOI:** 10.3390/s24144641

**Published:** 2024-07-17

**Authors:** Honglin Zhang, Jianhao Ye

**Affiliations:** School of Electronic and Information Engineering, South China University of Technology, No. 381 Wushan Avenue, Guangzhou 510641, China; lrqyjh@outlook.com

**Keywords:** wide bandwidth, patch antenna, high gain, field distribution, magnetic current

## Abstract

Communication systems need antennas with wide bandwidths to provide large throughput, while imaging radars benefit from high gain for increased range and wide bandwidths for high-resolution imaging. This paper presents the design and evaluation of a wideband, high-gain antenna that achieves an average gain of 9.7 dBi over a bandwidth of 1.49 GHz to 3.92 GHz by using multiple in-phase radiating apertures. The antenna has a unique structure with a central rectangular short-circuited patch sandwiched between two back-to-back U-shaped radiating patches and two flanking H-shaped short-circuited patches. Each of the U-shaped patches employs a coplanar waveguide as feeding to achieve ultra-wideband impedance matching. Benefiting from design arrangement, in-phase electrical field distributions appear at the gaps between the patches that result in equivalent radiating magnetic currents in the same direction. Theory analysis shows that the close-spaced, same-direction magnetic currents created by the radiating apertures intensify the radiation and increase antenna gain within its impedance bandwidth. Simulated data show that the use of the coplanar waveguide feeding and short-circuited patches increase the bandwidth from 65 MHz to 2.43 GHz. Moreover, the short-circuited patches increase the gain by 3.45 dB at 2.4 GHz. Simulation and measurement results validate the design and show that the antenna features a maximum gain of 11.3 dBi and an average gain of 9.7 dBi in a fractional bandwidth of 89.8%. Because of the high gain values and the wide bandwidth, the antenna is particularly suited for long-range communication systems and high-resolution radar applications.

## 1. Introduction

High-gain antennas capable of operating across wide bandwidths are pivotal for enhancing the performances of communication systems, enabling high-speed data transmission over long distances. These antennas are instrumental in radar technologies, facilitating the generation of high-resolution images from significant distances [1]. Although traveling-wave antennas, such as Vivaldi antennas, offer wide bandwidths, their end-fire radiation pattern limits their application, which requires antennas that radiate in the broadside direction [2]. Conversely, patch antennas are compact and radiate in the broadside direction but traditionally suffer from narrow bandwidths. Recent advancements have demonstrated that patch antennas achieved high gain over wide bandwidths while maintaining a compact form factor [3].

Numerous research efforts have been dedicated to the development of wideband, high-gain antennas for both communication and radar applications. Innovations have ranged from the use of L-shaped probes with distilled water to excite a transparent water patch, achieving a peak gain of 7.5 dBi within a 42.6% bandwidth [4], to the employment of parasitic patches and metasurfaces to enhance bandwidth and gain. For instance, a remodeled coplanar L-probe-fed patch antenna simplified the fabrication process while attaining a peak gain of 9.5 dBi and a 41% bandwidth [5]. Other notable designs include antennas with U-shaped slots, achieving an up to 68% bandwidth, and the use of grounded parasitic patches to realize gains of up to 10.5 dBi within specific bandwidths [6]. Incorporating periodic metal patterns and metasurfaces has proven effective in increasing antenna gain, with excellent designs achieving peak gains exceeding 13 dBi [7]. A miniaturized antenna in [8] used two layers of metasurfaces to increase the gain of a slot antenna to 6.48 dBi in a bandwidth of 44.43%. In [9], a wideband patch antenna using a metasurface increased the peak gain to 12.17 dBi in its impedance bandwidth of 65.06%. Parasitic structures are used to achieve high gain values, as shown in [10], where six grounded parasitic patches were used to achieve a 3 dB gain bandwidth of 46%, producing a peak gain of 10.5 dBi. The antenna using differential feeding and grounded parasitic patches in [11] achieved a peak gain about 10 dBi, but its bandwidth was only 26%. Reconstructing the reflecting plane and concentrating radiated power increases gain values, as shown in [12], which used two ground walls and increased its peak gain to 12 dBi in its 45% bandwidth. In [13], an ultra-wideband antenna used a square open cavity to increase the peak gain to 11.8 dBi, with an average gain of 6 dBi in a bandwidth of 117%. The antenna in [14] used a hollow cubic reflector to increase the gain of a bowtie patch’s peak gain to 10.3 dBi in its 128% fractional bandwidth. Our previous work in [15] used a hexagonal open cavity to increase the gain, achieving a peak gain of 12.9 dBi in a bandwidth of 113.3%.

Despite these advancements, the use of metasurfaces and reflective cavities, while beneficial for gain enhancement, might increase antenna height, presenting a challenge for applications requiring low-profile designs. Therefore, there remains a need for new approaches to achieve wideband, high-gain performance without compromising the compactness of the antenna.

This paper introduces a novel low-profile, high-gain antenna design that uses wideband feeding and in-phase radiating apertures to achieve improved gain performance in a wide bandwidth. The design achieves wide impedance bandwidth by using two embedded coplanar waveguide (CPW) feed lines and enhances gain performance by creating in-phase field distribution at the gaps between the patches. Thorough theory analysis shows that in-phase field distribution constructs close-spaced equivalent magnetic currents in the same direction, which intensifies the antenna’s radiation in the specified direction. Simulated and measured results confirm the antenna’s performance as they show a maximum gain of 11.3 dBi within an impedance bandwidth from 1.49 GHz to 3.92 GHz, which overlaps an 89.8% bandwidth of 3 dB gain variation. This antenna is a good choice for broadband communication and high-resolution radar systems, offering a versatile solution that combines broad bandwidth with high-gain performance in a compact form factor.

## 2. Materials and Methods

Figure 1 shows that the proposed antenna featured two U-shaped radiating patches, three grounded parasitic patches, two coplanar waveguide (CPW) feed lines, and three metallic shorting pins. Six plastic pillars were located near the edge of the substrate as spacers and fasteners. The radiating patches and CPW feed lines were etched on Rogers RO4003C laminates with a thickness of 0.8 mm, relative permittivity of 3.55, and tangential loss of 0.0027.

To achieve wide bandwidth, two identical CPW feed lines were used to connect the input ports to the two back-to-back U-shaped radiating patches. Three grounded parasitic patches were placed to create in-phase radiating apertures for high-gain radiation. Each of the grounded patches has a metallic shorting pin being soldered to the ground plane.

Figure 1b shows the dimension parameter labels of the antenna from its top and side views. It also shows the zoom-in image of the CPW lines from the side view. All dimension parameters and their final values are tabulated in Table 1.


**Impedance Bandwidth Enhancement**


Figure 2 illustrates two initial designs of the antenna. Antenna I in Figure 2a utilizes two narrow metal strips on a substrate as feed lines, whereas Antenna II in Figure 2b incorporates CPW lines printed on laminates for feeding.

The change in the feeding significantly improved bandwidth because of the impedance controlling effect brought by the CPW lines. The upper half of Figure 3 shows the real and imaginary parts of the two initial antennas. It shows that replacing the feeding lowered the high real impedance (Re. Z) between 1 GHz and 2.5 GHz to be close to 50 Ohms, which was the CPW line’s characteristic impedance. Similarly, the imaginary impedance Im.Z was tuned to be closer to 0 Ohms. The changes in Re.Z and Im.Z brought by the CPW lines led to the bandwidth improvement shown in the lower half of Figure 3. As indicated by the simulation results shown in Figure 3, Antenna I had poor matching because of its high impedance values below 2.5 GHz, while Antenna II possessed a fractional bandwidth of 73.5% (1.53–3.31 GHz) after replacing the thin metal strips with two CPW transmission lines with the same characteristic impedance of 50 Ohms.

Parametric analysis through dimension tuning revealed that the CPW line’s slot width (*s*) and length (*lc*) are critical for controlling impedance and the reflection coefficient. Figure 4, showing curves for different s values, suggests that an s value of 0.8 mm yielded the widest bandwidth by maintaining the impedance at around 50 Ohms. Similarly, Figure 5 displays curves for various $l_{c}$ values, indicating that $l_{c}$ = 55 mm maximized the bandwidth. Aiming at achieving a wide bandwidth, the optimal s and $l_{c}$ values were determined through optimization to be 0.8 mm and 54 mm, respectively, achieving an impedance bandwidth of 73.5% (1.53–3.31 GHz).


**Gain Drop and Its Improvement**


Although the CPW line enhanced bandwidth, Figure 6 reveals a substantial gain variation of 5.45 dB across the band, which complicated RF circuit design because of large gain differences.

Analysis shows that the gain drop at 2.4 GHz was caused by the weakened radiation intensity in the broadside direction. The relationship between antenna gain and radiation intensity is given as follows:(1)$$\begin{array}{ccc} G_{\max} & = & e_{t}D=e_{t}\frac{U_{\max}}{U_{0}}=e_{t}\frac{U_{\max}}{\frac{P_{rad}}{4\pi }}\\ & = & e_{t}4\pi \frac{U{|}_{\max}}{{∯}_{\Omega }Ud\Omega }=e_{t}4\pi \frac{|\overset{⃑}{E}{|}^{2}{|}_{\max}}{{∯}_{\Omega }|\overset{⃑}{E}{|}^{2}d\Omega } \end{array}$$

In (1), $e_{t}$ is the total radiation efficiency associated with matching and material losses; U is the radiation intensity, while $P_{rad}$ is the total radiated power; and $\overset{⃑}{E}$ is the electric field intensity at the observation point. The equation states that the gain relates to the field intensity, which is found through vector potential $\overset{⃑}{F}$ if magnetic current sources are known.

For an antenna with known magnetic currents, as shown in Figure 7, the corresponding vector potential $\overset{⃑}{F}$ and electric field $\overset{⃑}{E}$ are found with
(2)$$\begin{array}{ccc} \overset{⃑}{F} & = & \frac{\varepsilon }{4\pi }{∯}_{S^{\prime }}{\overset{⃑}{M}}_{s}\left(x^{\prime },y^{\prime },z^{\prime }\right)\frac{e^{-jkR}}{R}dS^{\prime }\\ & = & \sum_{i}{\overset{⃑}{F}}_{i}=\frac{\varepsilon }{4\pi }\sum_{i}\int_{Li}{\overset{⃑}{M}}_{si}\left(x^{\prime },y^{\prime },z^{\prime }\right)\frac{e^{-jkR_{i}}}{R_{i}}dl^{\prime } \end{array}$$
(3)$$\overset{⃑}{E}=-\frac{1}{\varepsilon }\nabla \times \overset{⃑}{F}$$

In (2), ε is the permittivity of the medium surrounding the antenna; k is the propagation constant in this medium; R or $R_{i}$ is the distance between the source and the observation point, which make them the function of both the source point $\left(x^{\prime },y^{\prime },z^{\prime }\right)$ and the observation point $\left(x,y,z\right)$; ${\overset{⃑}{F}}_{i}$ is the vector potential associated with the magnetic current ${\overset{⃑}{M}}_{si}$; and $L_{i}$ is the current path of ${\overset{⃑}{M}}_{si}$.

As the radiating structure is complex and not suitable for direct analysis, Huygens’s equivalence principle was applied to find the antenna’s equivalent magnetic currents. Figure 8 shows the Huygens box enclosing Antenna II and the electric field distributions on top of the box. The corresponding magnetic currents were plotted with dashed red arrows, as shown in Figure 8b. By putting a Huygens box infinitesimally close to the antenna and filling it with perfect electric conductor [16,17], the equivalent magnetic currents were found using ${\overset{⃑}{M}}_{s}=-2\overset{⃑}{n}\times {\overset{⃑}{E}}_{s}$ with the electric field ${\overset{⃑}{E}}_{s}$ appearing on the surface. The directions of the magnetic currents were upwards or downwards, in line with the *y*-axis negative and positive directions.

To reveal the relationship between field distribution and gain performance, field distributions at 1.6 GHz, 2.4 GHz, and 3.4 GHz are displayed in Figure 8b. The field distribution at 1.6 GHz shows that the electric field appeared to be uniform between the two U-shaped patches. Based on the field distribution and using Huygens’ principle, simplified and idealized equivalent magnetic currents were found and are drawn in Figure 8b. For later convenience, three equivalent magnetic currents with the same directions were assigned to the center gap. In comparison, the electric field in the same gap experiences direction change that results in three magnetic currents in opposite directions. Because the equivalent magnetic currents travel along the same direction parallel to the *y*-axis, Equation (1) is now rewritten as
(4)$$\overset{⃑}{F}=\frac{\varepsilon }{4\pi }\sum_{i}\int_{L_{i}}{\overset{⃑}{M}}_{si}\left(x^{\prime },y^{\prime },z^{\prime }\right)\frac{e^{-jkR_{i}}}{R_{i}}dl^{\prime }\approx \frac{\varepsilon }{4\pi }\int_{{y^{\prime }}_{\min}}^{{y^{\prime }}_{\max}}\sum_{i}\left({\overset{⃑}{M}}_{si}\left(x^{\prime },y^{\prime },z^{\prime }\right)\frac{e^{-jkR_{i}}}{R_{i}}\right)dy^{\prime }.$$

Equation (4) suggests that vector potential $\overset{⃑}{F}$’s amplitude and direction are associated with $\sum_{i}{\overset{⃑}{M}}_{si}\left(y^{\prime }\right)\frac{e^{-jkR_{i}}}{R_{i}}$, which is the sum of multiple complex exponential terms. By applying complex analysis principles and letting $m_{i}=\left|\left.{\overset{⃑}{M}}_{si}\left(x^{\prime },y^{\prime },z^{\prime }\right)\right|\right./R_{i}$ and $\psi_{i}=-kR_{i}$, (4) is re-written as
(5)$$\overset{⃑}{F}\approx \frac{\varepsilon }{4\pi }\int_{{y^{\prime }}_{\min}}^{{y^{\prime }}_{\max}}{\overset{⃑}{a}}_{y}Me^{j\Psi }dy^{\prime }$$
where M and Ψ are found with
(6)$$M=\sqrt{\sum_{i}{\left|m_{i}\right|}^{2}\pm \sum_{1\leq i<j\leq n}2|m_{i}||m_{j}|\cos\left(\psi_{j}-\psi_{i}\right)},$$
(7)$$\Psi =\arctan\left(\frac{\sum_{i}m_{i}\sin\left(\psi_{i}\right)}{\sum_{i}m_{i}\cos\left(\psi_{i}\right)}\right).$$

In Equation (5), a plus sigh is taken if ${\overset{⃑}{M}}_{si}$ and ${\overset{⃑}{M}}_{sj}$ have the same direction, namely they are in phase. Equations (5) and (6) suggest that the magnitude of $\overset{⃑}{F}$ decreases when $\cos\left(\psi_{j}-\psi_{i}\right)$ takes a negative value, which corresponds to $\frac{\pi }{2}<\left(\psi_{j}-\psi_{i}\right)<\frac{3\pi }{2}$. By replacing $\psi_{i}=-kR_{i}$ and k=2π/λ, it was deduced using the equations that the magnitude of $\overset{⃑}{F}$ decreases if the difference of distance $R_{i}$ and $R_{j}$ satisfies
(8)$$\frac{\lambda }{4}<R_{i}-R_{j}<\frac{3\lambda }{4}.$$

When $R_{i}$ and $R_{j}$ meet the condition given by Equation (8), the magnitude of $\overset{⃑}{F}$ decreases even ${\overset{⃑}{M}}_{si}$ and ${\overset{⃑}{M}}_{sj}$ are of the same direction, leading to a reduced gain value. If the distance difference $\Delta R_{ij}=|R_{i}-R_{j}|$ is smaller than 1/4 wavelength or larger than 3/4 wavelength, the radiation intensity is increased. It should be noted that the observation point should not be set to the infinite or else $R_{i}$ equals $R_{j}$, and the locations of the magnetic currents have no influence on the magnitude of vector potential $\overset{⃑}{F}$ and antenna gain. 

When $\Delta R_{ij}=|R_{i}-R_{j}|$ is so small that $\Delta R_{ij}\ll \lambda /20$ and $k\Delta R_{ij}\ll \pi /10$, the phase offset brought by $e^{-jkR_{i}}$ with different $R_{i}$ is negligible. Thus, the distances from the sources to the observation point are about the same, namely $R_{i}\approx R_{j}\approx R$. Here, Equation (4) becomes
(9)$$\overset{⃑}{F}=\frac{\varepsilon }{4\pi }\sum_{i}\int_{L_{i}}{\overset{⃑}{M}}_{si}\left(x^{\prime },y^{\prime },z^{\prime }\right)\frac{e^{-jkR_{i}}}{R_{i}}dl^{\prime }\approx \frac{\varepsilon }{4\pi }\int_{y_{\min}}^{y_{\max}}\frac{e^{-jkR}}{R}\sum_{i}{\overset{⃑}{M}}_{si}\left(y^{\prime }\right)dy^{\prime }.$$

Equation (9) suggest that vector potential $\overset{⃑}{F}$ is the function of the vector sum of magnetic currents if they are closely spaced. If the magnetic currents take the same direction, $\overset{⃑}{F}$ achieves its maximum value. 

Using (9) and knowing the directions and locations of the magnetic currents shown in Figure 8b, it is clear that $\overset{⃑}{F}$’s magnitude is decreased at 2.4 GHz since ${\overset{⃑}{M}}_{s0}$ takes an opposite direction as those of ${\overset{⃑}{M}}_{s1}$, ${\overset{⃑}{M}}_{s2}$, ${\overset{⃑}{M}}_{s1}$, and ${\overset{⃑}{M}}_{s1}$, while the magnetic currents near the center have the same direction at 1.6 GHz, leading to a high gain value at this frequency. The decreased $\left|\overset{⃑}{F}\right|$ thus leads to a reduction in gain at 2.4 GHz, as shown by the simulated gain curve shown in Figure 6. 

To increase the gain at 2.4 GHz, the center equivalent magnetic current ${\overset{⃑}{M}}_{s0}$ must be removed or redirected to produce an in-phase distribution. Analysis on simulated data shows that adding a shorted patch at the center prevents the electric field from radiating and significantly decreases the magnitude of ${\overset{⃑}{M}}_{s0}$, as shown in Figure 9. The field intensity at the center of Antenna III is decreased at 2.4 GHz, leading to a reduction in $\left|{\overset{⃑}{M}}_{s0}\right|$. According to (9), the reduction in $\left|{\overset{⃑}{M}}_{s0}\right|$ increases the magnitude of vector potential $\overset{⃑}{F}$, which results an enhanced gain value at 2.4 GHz. 

Antenna III’s gain was further improved by adding two H-shaped shorting patches, which constructed Antenna IV in the manner shown in Figure 10. The gain enhancement was achieved by introducing in-phase field distribution beyond 2.25 GHz. From the field distribution at 2.4 GHz and 3.4 GHz shown in Figure 10, strong and in-phase electric field appeared at the inner edges of the H-shaped patches, resulting simplified equivalent magnetic currents (${\overset{⃑}{M}}_{s5}$ to ${\overset{⃑}{M}}_{s10}$) with preferred directions. Because of the in-phase field distribution, the shorting patches increased the antenna’s gain from 2.25 GHz to 3.5 GHz, according to the result curves shown in Figure 11. 

Simulated *S*-parameter curves in Figure 11 show that the center shorting patch increased the antenna’s impedance bandwidth as it improved the matching beyond 2.4 GHz. Bandwidth improvement was also observed after adding the two H-shaped shorting patches, as the lower operating frequency was moved from 1.5 GHz to 1.38 GHz. The simulated data related to the curves in Figure 3 and Figure 11 indicate that the simulated bandwidths of Antenna I, Antenna II, Antenna III, and Antenna IV are 65 MHz (from 1.537 GHz to 1.602 GHz), 1.78 GHz (from 1.53 GHz to 3.31 GHz), larger than 2.3 GHz (from 1.47 to over 4 GHz), and larger than 2.6 GHz (from 1.4 GHz to over 4 GHz), respectively.

The gain improvement is reflected by the radiation patterns shown in Figure 12, which shows that Antenna IV had the smallest side lobes on both E- and H-planes at 2.4 GHz among the three antennas. The pattern curves in Figure 12 verify that the shorting patches increased the antenna’s gain by boosting the main level with reduced side lobe levels.

Parametric analysis through dimension tuning shows that two dimensional parameters relating to the H-shaped patches played vital roles regarding gain performance. It should be noted that other parameters were unchanged while tuning the dimension under discussion.

Figure 13 shows that changing the widths ($w_{4}$) of the horizontal strip in the H-shaped patches affected the gain values beyond 1.75 GHz. When $w_{4}$ was reduced from 24 mm to 4 mm, the gain at 2.2 GHz increased from 5.51 dBi to 10.57 dBi. 

Parametric analysis also showed that the lengths (*l*_3_) of the inner vertical arms of the H-shaped patches affected the gain values near 2.2 GHz, as shown by the curves in Figure 14. They show that when *l*_3_ was increased from 20 mm to 60 mm, the gain at 2.2 GHz was increased from 8.12 dBi to 10.47 dBi. 

## 3. Results

The antenna is fabricated and tested with an assembled wideband balun because Antenna IV needs a circuit to provide signals with equal amplitudes and reversed phase. The balun has the same design as in [15,18,19]. The antenna and the balun circuit are designed separately and then joined to form an assembled 3D model. The balun’s structure and the performance regarding input port return loss and output ports balance are shown in Figure 15. It shows that the balun provides good balance between the output ports since the amplitude and phase differences are less than 1.7 dB and 9 degrees, respectively.

To reduce height and cost, the antenna uses the bottom metallic sheet as its ground plane. The balun is printed on both sides of a dielectric laminate with a thickness of 0.508 mm, relative permittivity of 3.55, and tangential loss of 0.0027. Through optimization, the antenna’s final dimension values are found and tabulated in Table 1. A full-wave simulation is performed to find the simulation results. Figure 16 shows the 3D model and photo of the assembled antenna.

Figure 17 shows the simulated and measured reflection coefficient curves of the assembled antenna. The measured results show that the assembled antenna, including the balun, has an absolute bandwidth of 1.49–3.9 GHz and a fractional bandwidth of 89.8%. Figure 17 shows that the measured bandwidth (|*S*_11_| < −10 dB) is slightly larger than the simulated one. Based on our analysis, the difference in $\left|S_{11}\right|$ is mainly caused by the assembling of the antenna. During the fabrication, an SMA connector and three thin metallic rods were soldered by hand. The difference between the soldering and the 3D model of the antenna leads to small errors. Besides the soldering, the difference is also the result of a slightly bent ground plane of the antenna. The ground plane is a 0.035 mm thick metallic sheet printed on a thin dielectric substrate, which is also used to fabricate the balun. Because the substrate’s thickness is only 0.508 mm, it bent slightly during the assembling of the antenna. Simulated and measured data show that the measured bandwidth is from 1.49 to 3.92 GHz, that is, 130 MHz larger than the simulated one, which is from 1.5 GHz to 3.8 GHz. Other than that, the measured $\left|S_{11}\right|$ agrees well with the simulated one.

Figure 18 shows the simulated and measured radiation efficiency and gain of the antenna. The figure shows a measured gain of 8.7–11.3 dBi and a radiation efficiency above 70% within the bandwidth from 1.49 to 3.5 GHz. The figure shows that the measured radiation efficiency is smaller than the simulated one. The reason for the discrepancy might be that there are differences related to materials between the fabricated antenna and its 3D model. Since the total efficiency $e_{t}=e_{r}e_{c}e_{d}$ is affected by matching ($e_{r}$) and material losses (conduction efficiency $e_{c}$ and dielectric efficiency $e_{d}$) [17], there are two causes of the reduction in the measured efficiency, namely matching or material losses. Because the simulated and measured |*S*_11_| curves shown in Figure 17 agree well with each other, matching could not be the real reason for the efficiency drop. Therefore, the real reason for the reduced radiation efficiency is that the material losses in the fabricated antenna are larger than those in the 3D model. Because of the reduction in measured radiation efficiency, the measured realized gain is also smaller than the simulated one, as indicated by the gain curves shown in Figure 18.

Figure 19 shows the simulated and measured radiation patterns on the E- and H-planes at 1.5 GHz, 2.5 GHz, and 3.5 GHz, which are at the lower boundary, middle, and upper boundary of the bandwidth. The main lobes at the three frequencies direct towards the broadside, showing that the antenna has good radiation in the considered direction. The figure shows that the side lobe’s level is over 12 dB, and the front-to-back ratio is over 15 dB at all frequencies.

## 4. Discussion

Table 2 compares this antenna to other antennas working in similar frequency bands, where *f_l_* is lowest operating frequency and λ*_l_* the corresponding wavelength. The proposed antenna has the largest impedance and 3 dB gain bandwidth, while it attains a high peak gain and small size.

## 5. Conclusions

This paper presents a low-profile patch antenna with a maximum gain of 11.3 dBi and an 89.8% of fractional bandwidth. It provides the methods used to improve impedance bandwidth and gain performance. Theory analysis shows that the gain improvement is brought by in-phase field distribution occurring at the gaps between various patches. The test of a fabricated prototype validates the performance improvements. Measured results agree well with simulated ones and show that the antenna radiates towards the broadside direction with an average gain of 9.7 dBi and radiation efficiency of over 70% in the frequency band from 1.49 GHz to 3.92 GHz. The antenna is ideal for communication, radiolocation, and radar applications, as it features a high average gain in a broad bandwidth.

## Figures and Tables

**Figure 1 sensors-24-04641-f001:**
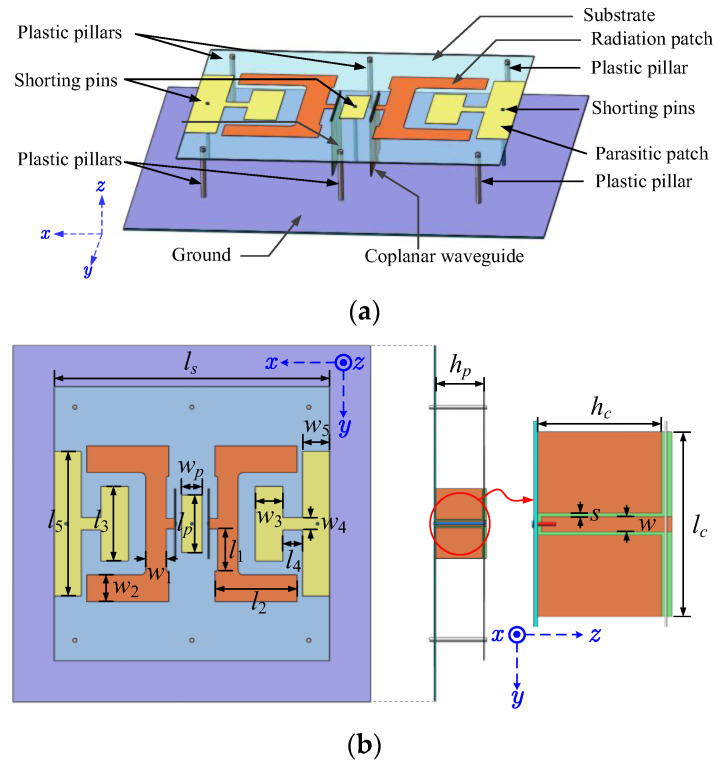
Antenna configuration: (**a**) perspective view; (**b**) top and side view.

**Figure 2 sensors-24-04641-f002:**
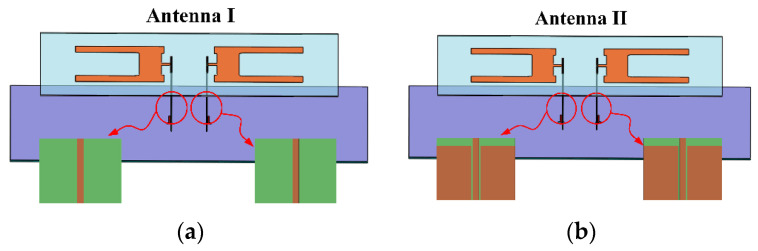
Initial stages of the antenna: (**a**) Antenna I fed by metal strips; (**b**) Antenna II fed by CPW lines.

**Figure 3 sensors-24-04641-f003:**
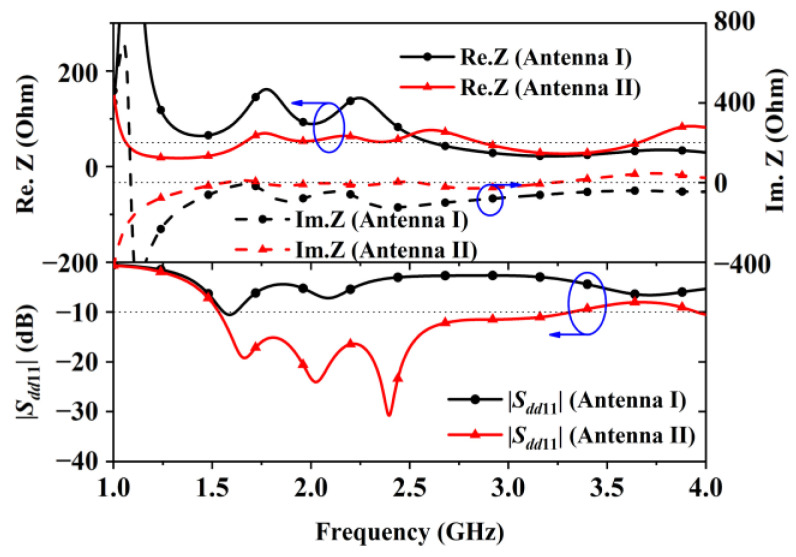
Reflection coefficients and impedance curves of Antenna I and Antenna II.

**Figure 4 sensors-24-04641-f004:**
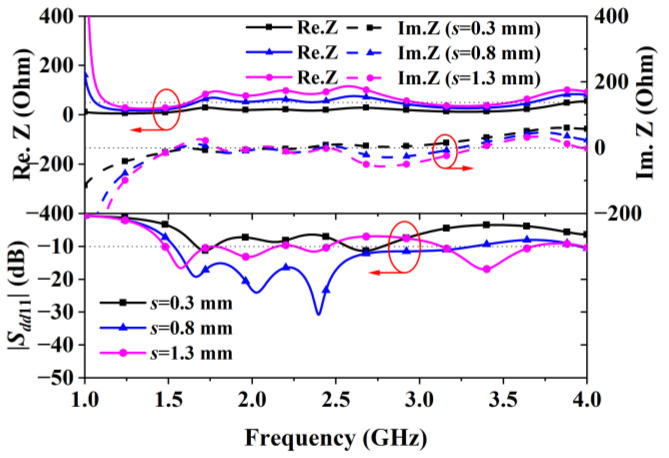
Impedance and reflection coefficient of Antenna II obtained with different s values.

**Figure 5 sensors-24-04641-f005:**
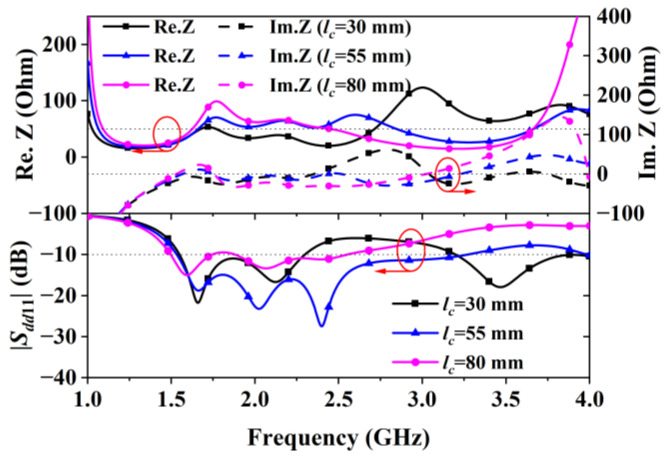
Impedance and reflection coefficient of Antenna II obtained with different $l_{c}$ values.

**Figure 6 sensors-24-04641-f006:**
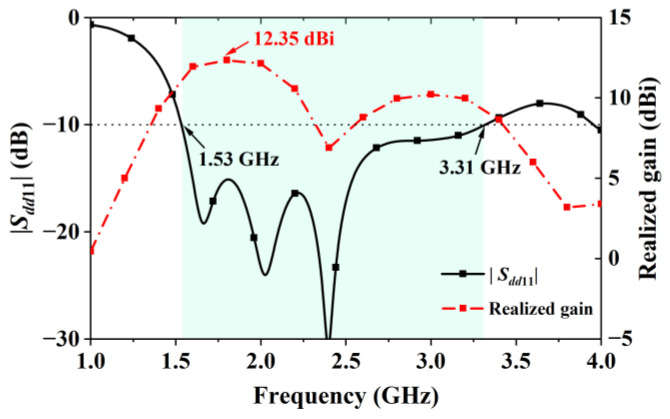
Reflection coefficient and gain curves of Antenna II.

**Figure 7 sensors-24-04641-f007:**
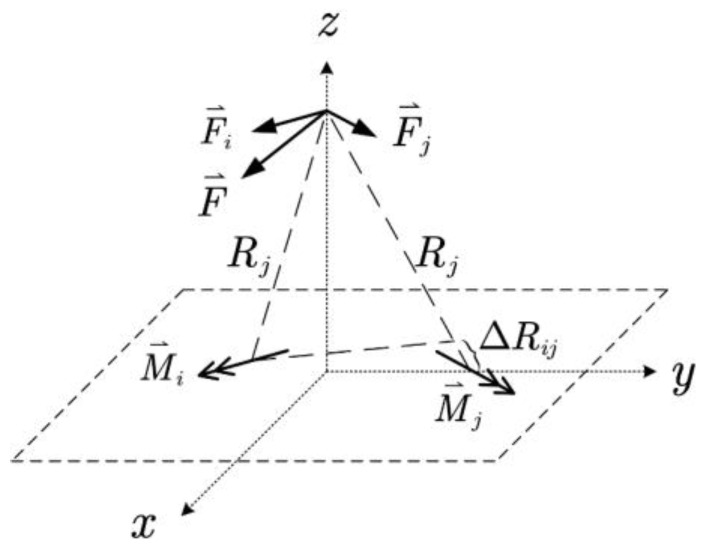
Vector potential and its magnetic current source.

**Figure 8 sensors-24-04641-f008:**
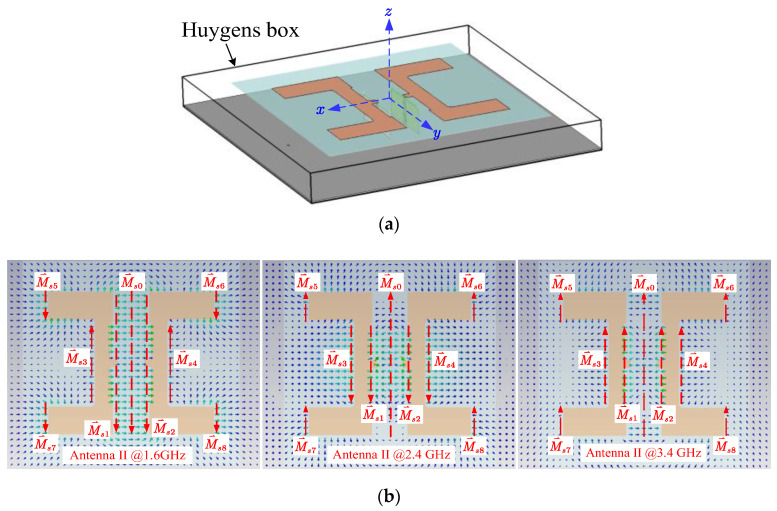
Huygens box and field distributions on the top side of the box: (**a**) the Huygens box surrounding Antenna II; (**b**) field distributions and their associated magnetic currents of Antenna II at 1.6 GHz (**left**), 2.4 GHz (**middle**), and 3.4 GHz (**right**).

**Figure 9 sensors-24-04641-f009:**
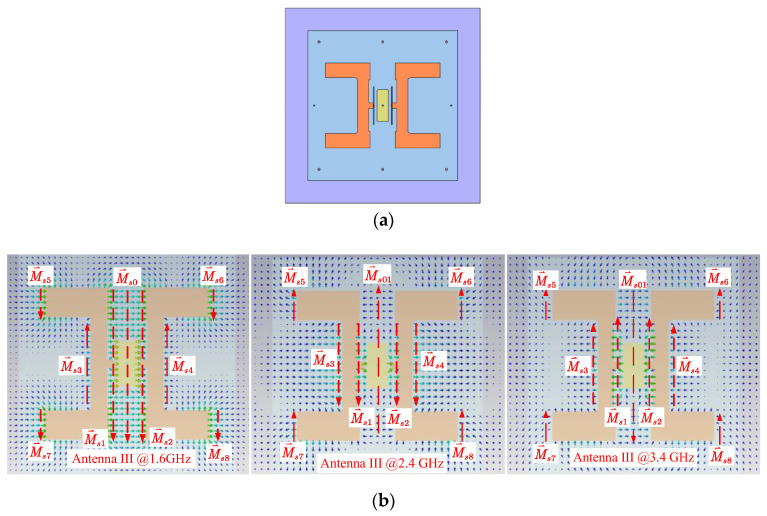
Top view of Antenna III and its field distributions. (**a**) Top view of Antenna III. (**b**) Field distributions at 1.6 GHz (**left**), 2.4 GHz (**middle**), and 3.4 GHz (**right**).

**Figure 10 sensors-24-04641-f010:**
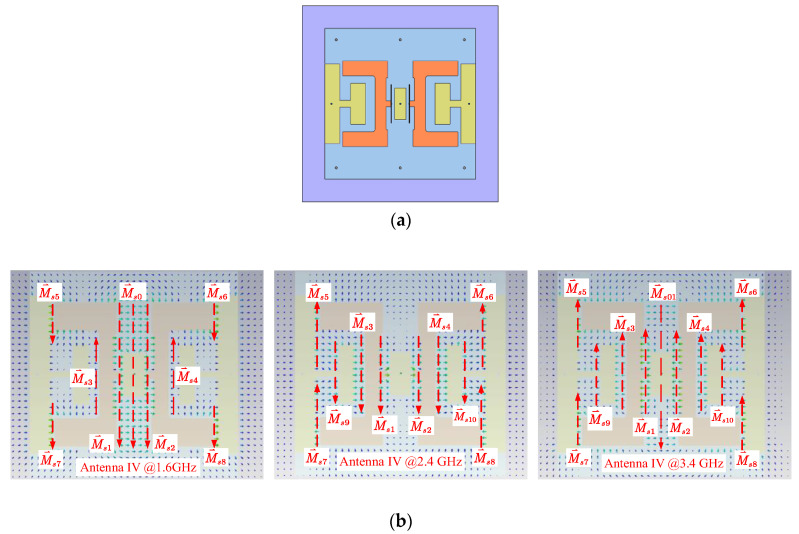
Top view of Antenna IV and its field distributions. (**a**) Top view of Antenna III. (**b**) Field distributions at 1.6 GHz (**left**), 2.4 GHz (**middle**), and 3.4 GHz (**right**).

**Figure 11 sensors-24-04641-f011:**
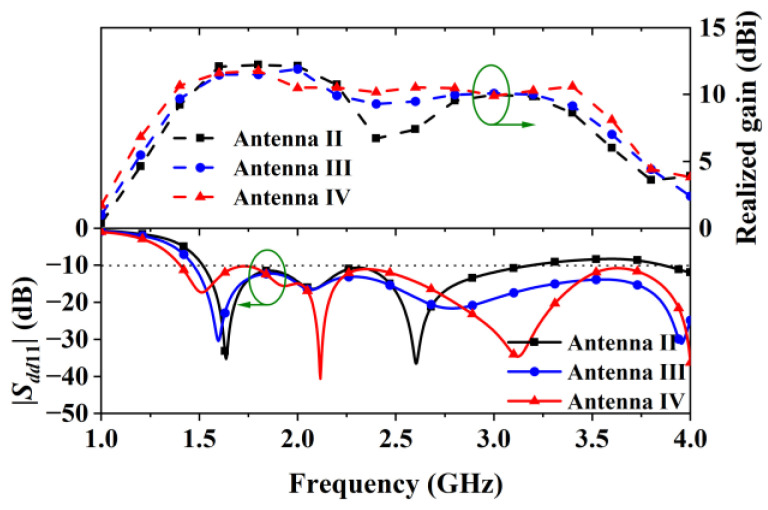
Reflection coefficients and gains of Antenna II, Antenna III and Antenna IV.

**Figure 12 sensors-24-04641-f012:**
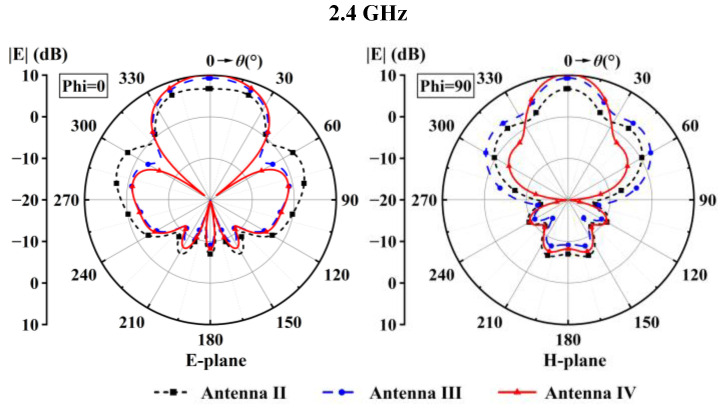
Radiation patterns of Antenna II, Antenna III, and Antenna IV.

**Figure 13 sensors-24-04641-f013:**
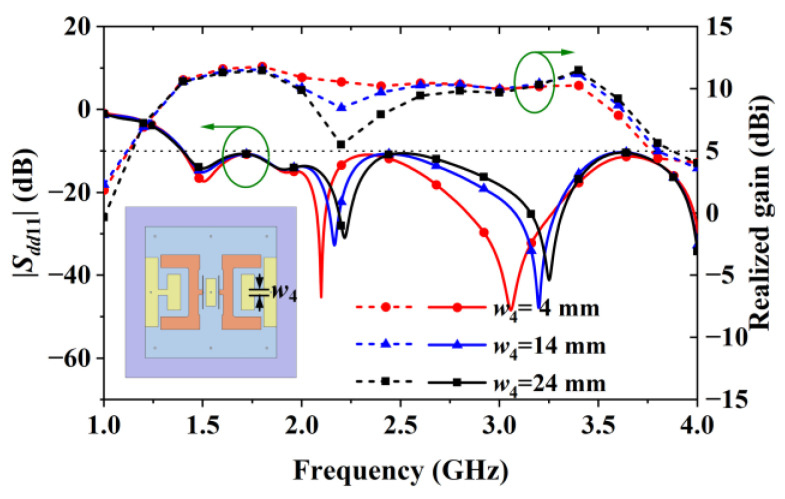
Reflection coefficient and gain of Antenna IV obtained with different $w_{4}$ values.

**Figure 14 sensors-24-04641-f014:**
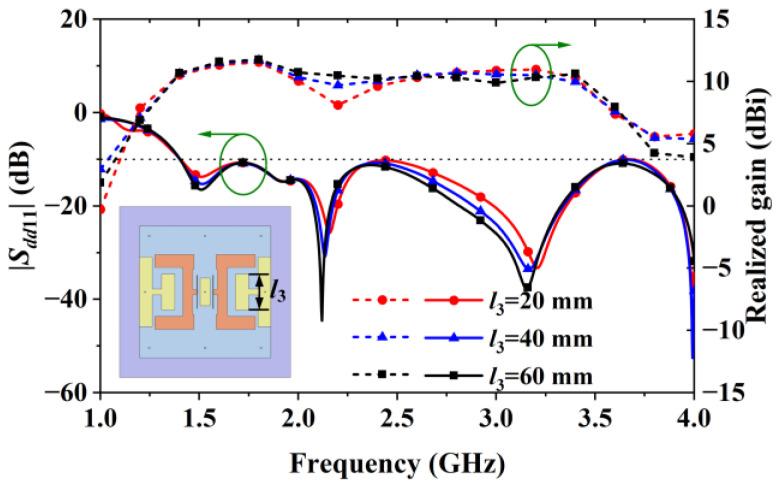
Reflection coefficient and gain of Antenna IV obtained with different *l_3_* values.

**Figure 15 sensors-24-04641-f015:**
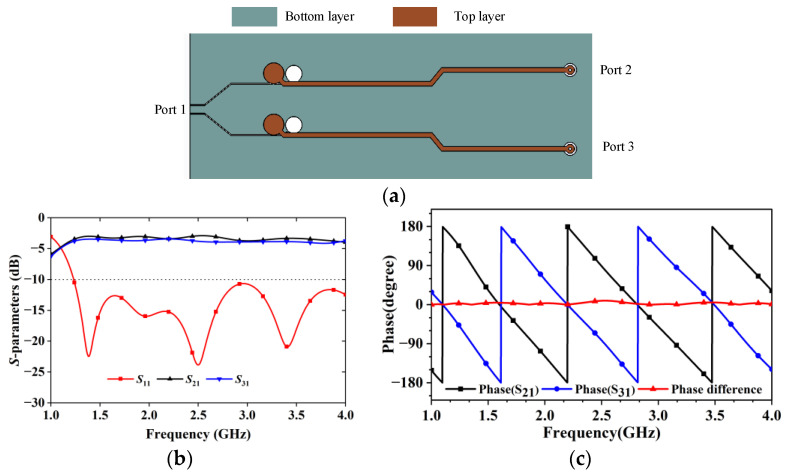
The feeding balun’s structure and its performance: (**a**) the balun’s structure and layer configurations, (**b**) amplitudes at the balun’s ports, and (**c**) phases at the balun’s output ports.

**Figure 16 sensors-24-04641-f016:**
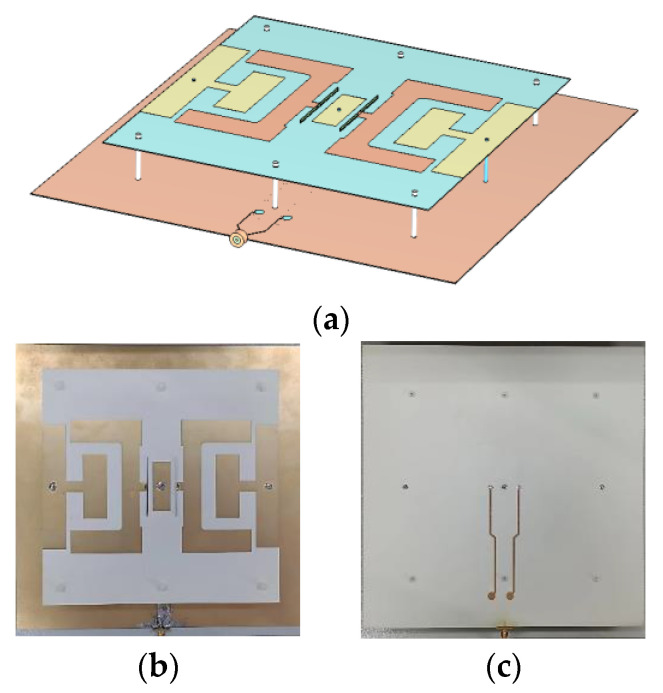
Three-dimensional model and photograph of the assembled antenna: (**a**) 3D model of the assembled antenna, (**b**) front view, and (**c**) back view.

**Figure 17 sensors-24-04641-f017:**
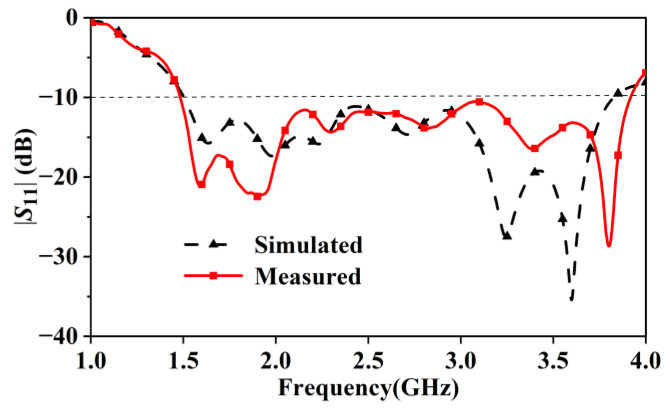
Simulated and measured reflection coefficients of the assembled antenna with a wideband balun.

**Figure 18 sensors-24-04641-f018:**
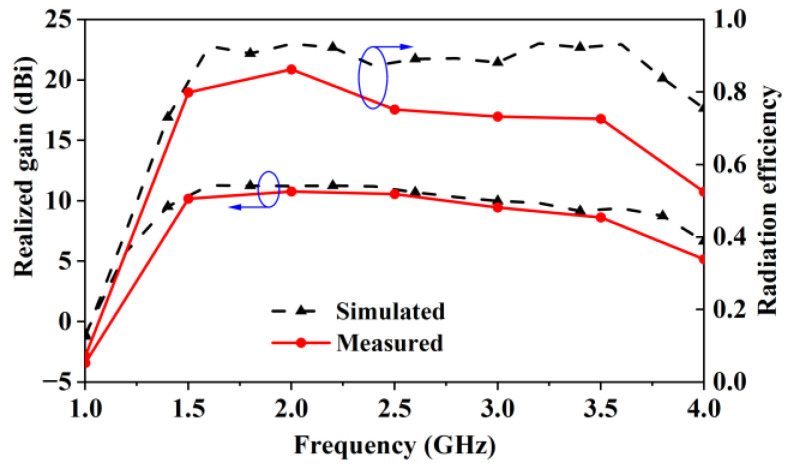
Simulated and measured realized gain and radiation efficiency of the assembled antenna.

**Figure 19 sensors-24-04641-f019:**
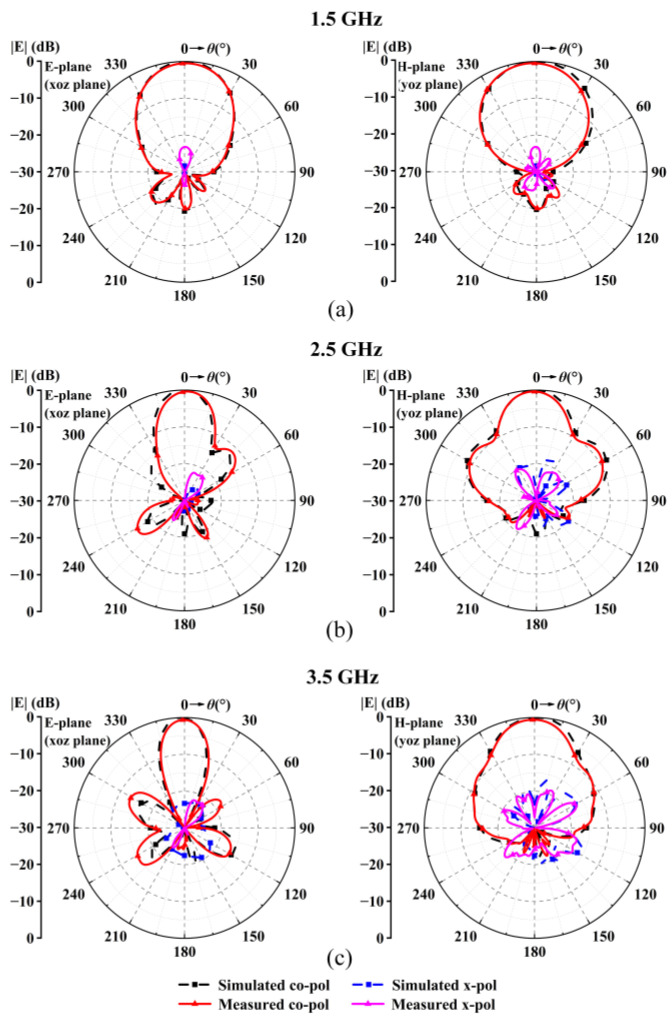
Radiation patterns of the proposed antenna on the E and H planes. (**a**) 1.5 GHz. (**b**) 2.5 GHz. (**c**) 3.5 GHz.

**Table 1 sensors-24-04641-t001:** Dimensions of the proposed antenna.

Parameters	*l* _1_	*l* _2_	*l* _3_	*l* _5_	*l_p_*	*w* _1_	*w* _2_	*w* _3_
Values (mm)	35.1	59.6	57.9	153.3	42	14.9	27.2	15
Parameters	*w* _5_	*w_p_*	*l_s_*	*h_c_*	*s*	*w*	*l_c_*	*h_p_*
Values (mm)	20.4	20	200	21.7	0.8	2.8	54	25

**Table 2 sensors-24-04641-t002:** Antenna’s performances compared with those of previously reported antennas.

Ref.	BW (|*S*_11_| < −10 dB)	3 dB Gain Bandwidth	Peak Gain (dBi)	Size (λ*_l_*^3^)
[4]	1.48–2.28 (42.6%)	1.5–2.24 (39.6%)	7.5	1.07 × 1.07 × 0.17
[11]	2.03–2.65 (26.5%)	2.1–2.65 (23.2%)	10.5	0.58 × 0.64 × 0.04
[12]	3.27–5.15 (44.6%)	3.3–4.54 (31.6%)	12.2	1.09 × 1.16 × 0.28
This work	1.49–3.92 (89.8%)	1.49–3.92 (89.8%)	11.3	0.99 × 0.99 × 0.12

## Data Availability

Data are contained within this article.

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
