# Peer review of "A Wideband High Gain Differential Patch Antenna Featuring In-Phase Radiating Apertures"

_sensors, 2024, doi:10.3390/s24144641_

Round 1

Reviewer 1 Report

Comments and Suggestions for Authors

The manuscript mainly proposed a composite structure to improve the maximum gain and impedance matching through the CPW feeding lines, parasitic patches, and shorting pins. The design is novel, but there are still some points need to be improved or explained as follows:

1. For Lines from 20-21, could the authors tell the future readers, how much percentage for gain or bandwidth, the improvements from Ant1, Ant2, Ant3, to Ant4, step by step? Of course we know that if use the CPW from balance to imbalance transformation would extent the bandwidth. Why not directly tell the improvement from Ant 2 to Ant 4?

2. In Fig. 1 (a), the six plastic pillars located did not depict out;

3. For Lines from 126 to 127, the optimal s and lc are 0.8 mm and 55 mm, which is different from the ones shown in Tab I, where, lc = 54 mm;

4. The analysis related with equivalent magnetic currents are too ideal as drawn in Fig. 8-10, which may not right to the real fields solutions. Just as mentioned in Line 157-159, the radiating structure is complex and not suitable for direct analysis. I am afraid I would not agree with this extra explanation. Actually, the equivalent magnetic current would follow the slot direction and the design structure slot is not ideal one;

5. In Section 3 Results, the preferred analysis is to do the circuit-field co-simulation, which probably would attract the readers more attention;

6. Finally, could the authors explain why in Fig. 14 and 15, there seem much difference for the higher frequency band |S11| and Radiation Efficiency between the simulated and measured results?

Comments on the Quality of English Language

Some places need to be improved for academic norms:

1. The unit for antenna gain is "dBi", but some place like Line 13, 60, etc., use "dB";

2. Please use "|S11|" to represent the value of reflection coefficients as in Fig. 14 (please check the whole manuscript);

3. Please note the plural form of the verb, like in Line 57, "increases" should be “increase”.

Author Response

Dear respected reviewer,

Thank you for your time and patience in reviewing our manuscript. And thank you for the invaluable comments and suggestions. We have prepared a point-to-point response document as the reply to your comments. Please see the attached file. 

Best regards,

Honglin Zhang

Reviewer 2 Report

Comments and Suggestions for Authors

The originality of the wideband antenna design in this manuscript is unquestionable. The design process is clearly outlined and backed by theoretical analysis and electromagnetic simulations. Experimental results are in good agreement with simulations. The article is structured logically. 

There are several minor remarks:

1. line 261. I suppose there should be a reference to Fig.13 instead of Fig.11.

2. The CPW was chosen by the authors and they provided a thorough explanation for their decision. Nonetheless, the design of the assembled antenna appears to be unreliable due to the CPWs acting as a "backbone". The article would be improved by including some thoughts on alternative feed lines. 

Author Response

(The authors gave the same response as above.)
